# From the Bench to the Bedside: Branched Amino Acid and Micronutrient Strategies to Improve Mitochondrial Dysfunction Leading to Sarcopenia

**DOI:** 10.3390/nu14030483

**Published:** 2022-01-22

**Authors:** Mario Romani, Mette M. Berger, Patrizia D’Amelio

**Affiliations:** 1Aging and Bone Metabolism Laboratory, Department of Medicine, University of Lausanne Hospital (CHUV), Service of Geriatric Medicine & Geriatric Rehabilitation, Mont-Paisible 16, 1011 Lausanne, Switzerland; Mario.Romani@chuv.ch; 2School of Medicine, University of Lausanne Hospital, 1011 Lausanne, Switzerland; Mette.Berger@chuv.ch

**Keywords:** sarcopenia, malnutrition, mitochondria, inflammageing, senescence, vitamin D, branched-chain amino acids (BCAAs), *n*-3 PUFA, zinc, selenium

## Abstract

With extended life expectancy, the older population is constantly increasing, and consequently, so too is the prevalence of age-related disorders. Sarcopenia, the pathological age-related loss of muscle mass and function; and malnutrition, the imbalance in nutrient intake and resultant energy production, are both commonly occurring conditions in old adults. Altered nutrition plays a crucial role in the onset of sarcopenia, and both these disorders are associated with detrimental consequences for patients (e.g., frailty, morbidity, and mortality) and society (e.g., healthcare costs). Importantly, sarcopenia and malnutrition also share critical molecular alterations, such as mitochondrial dysfunction, increased oxidative stress, and a chronic state of low grade and sterile inflammation, defined as inflammageing. Given the connection between malnutrition and sarcopenia, nutritional interventions capable of affecting mitochondrial health and correcting inflammageing are emerging as possible strategies to target sarcopenia. Here, we discuss mitochondrial dysfunction, oxidative stress, and inflammageing as key features leading to sarcopenia. Moreover, we examine the effects of some branched amino acids, omega-3 PUFA, and selected micronutrients on these pathways, and their potential role in modulating sarcopenia, warranting further clinical investigation.

## 1. Introduction

The life expectancy of an average person has doubled during the last century; hence, the elderly population is rapidly increasing. Disability due to age-related diseases and frailty has become an important socio-economic burden. Amongst chronic conditions that severely affect the patient’s quality of life, sarcopenia is widely present, but generally underdiagnosed, and therefore undertreated. Sarcopenia and its consequences are essential features of “physical frailty”, and contribute to “unhealthy aging”. The term sarcopenia indicates a condition characterized by reduced muscle mass and muscle strength (“Dynapenia”), leading to an impairment in physical performance [1]. Sarcopenia is associated with poor health outcomes, such as functional decline, falls, fractures, depression, and mortality [1], which increases health care costs and decreases quality of life in older patients [2,3,4]. A contributing factor to sarcopenia under-diagnosis is the diversity of diagnostic criteria; recently, these criteria have evolved (Table 1). The more ancient criteria are essentially based on the reduction of muscle mass regardless of its effect on muscle strength and performance [5,6], whereas the more recent criteria are stricter [7,8,9].

The European Working Group on Sarcopenia (EWSGOP) has proposed an evolution of their former algorithm (EWGOP1) [8]; hence, it is defined as EWSGOP2 criteria [10]. The new criteria suggest using the SARC-F questionnaire as a screening tool to identify patients at risk [11], to measure muscle weakness as the primary characteristic of sarcopenia, and to confirm the diagnosis by measuring muscle quantity or quality (Table 1 and Figure 1). EWGSOP2 further recommends the use of grip strength [12] and a chair stand test [13] to measure muscle strength. The Double-emission X-ray Absorptiometry (DXA) and the Bioimpedentiometry Analyses (BIA) are advocated for muscle mass and quality measurement in standard clinical care, whereas the use of MRI or CT is recommended only for research or for special needs in patients at high risk of adverse clinical outcomes. In order to assess the presence of severe sarcopenia, the EWSGOP-2 advises to measure physical performance by using the Short Performance Physical Battery (SPPB) [14], the Timed “Up and Go” test (TUG) [15], and the 400-m walk tests [16] (Table 1).

A recent study compared EWSGOP-2 and EWSGOP-1 criteria in in two cohorts of Swedish older adults, showing small differences in the prevalence of sarcopenia measured either way (0.9–1.0 percentage points lower with EWSGOP-2). However, in the very old subjects (>85-years), significant differences between EWGSOP-1 and 2 could not be ruled out [17]. The prevalence of the disease ranges from 6.42% to 21.56% depending on the diagnostic criteria applied and the population studied [18]. Regardless, the number of patients living with sarcopenia will grow to over 200 million in the next 40 years worldwide [19]. Moreover, despite the criteria used and population analyzed, sarcopenia is associated with a significantly higher mortality (HR: 2.00 (95% CI: 1.71, 2.34); OR: 2.35 (95% CI: 1.64, 3.37)) [20].

Several risk factors have been associated with the development of sarcopenia; among those, aging [21,22], malnutrition [23,24], and reduced physical activity [25] play an important role. Despite the epidemiology and the clinical consequences of sarcopenia, until now, no validated treatment has emerged, although several drugs are under investigation [26]. Current clinical practice relies on a rehabilitative approach that includes physical exercise and nutritional interventions [27,28].

The diagnosis and treatment of sarcopenia are further complicated by the lack of reliable markers of muscle health and targets for potential interventions. Different biological pathways have been proposed to discriminate between physiological ageing and pathological ageing, named “senescence” [29]. Cellular senescence has been shown to be responsible for reduced muscle mass and quality observed in sarcopenia; namely the senescence of satellite cells, the development of a senescence-associated secretory phenotype, and an imbalance in protein synthesis/degradation and inflammation have been shown to be of paramount importance in the pathogenesis of sarcopenia [30,31]. Amongst the multiple mechanisms involved in cellular senescence, here we focus on mitochondrial dysfunction as associated to increased oxidative stress and inflammation, and their role in the development of sarcopenia. Mitochondrial dysfunction and increased oxidative stress are of particular interest in the development of age-associated sarcopenia, as they are potentially reversible with a nutritional intervention [32]. Lifestyle, physical activity, and dietary intake, which greatly influence the development of sarcopenia, are implicated in the regulation of mitochondrial function, and oxidative stress as well [22,32,33,34].

Numerous nutrients have been linked both to mitochondrial function and to muscle health [35]. Among macronutrients, the branched-chain amino acids (BCAAs) [36] and the omega 3 polyunsaturated fatty acids (*n*-3 PUFA) [37] have repeatedly been shown to have a central place. Among micronutrients, vitamin D, zinc, and selenium, which all have antioxidant properties, have been shown to be essential. The status of these nutrients has been shown to be inadequate or deficient in a large proportion of old adults.

Although several reviews investigated the role for nutritional interventions in rescuing sarcopenia, here we propose an original translational approach unravelling the link between mitochondrial dysfunction, oxidative stress, chronic inflammation, and ageing with their impact on sarcopenia, highlighting how a nutritional strategy may be optimized to mitigate cellular senescence and, therefore, sarcopenia at clinical level.

## 2. Mitochondrial Dysfunction and Oxidative Stress

Abundant literature has described an age-dependent impairment of mitochondrial function, and an alteration of mitochondrial structure across species from yeast to humans, which in turn contributes to age-associated cellular quality control decline and tissue dysfunction [38,39]. Though dysregulation of mitochondrial homeostasis is a global hallmark of aging, there are tissues in which it is more prominently observed. These typically include post-mitotic and highly metabolic tissues, such as skeletal muscle, which are therefore more sensitive to the dysregulation of mitochondrial-mediated processes [38,40]. Hence, it is not surprising that altered mitochondrial homeostasis has been long suggested as key player in the development of sarcopenia.

Indeed, sarcopenic muscles present reduced mitochondrial mass, largely due to decreased mitochondrial biogenesis (Figure 2). Mitochondria biosynthesis relies on the key transcription factor, peroxisome proliferator-activated receptor gamma coactivator 1-alpha (PCG1α). In agreement with the observed age-associated mitochondrial content reduction and dysfunction, PGC1-α mRNA and protein levels are reduced in aged experimental animals and older persons [41,42,43,44,45,46]. Interestingly, PGC1α overexpression is sufficient to prevent age-associated muscle dysfunction and atrophy, contributing to maintain mitochondrial copy number, and normalize mitochondrial dynamics [47,48,49]. Moreover, PCG1α modulates the expression of a panel of antioxidant enzymes, including superoxide dismutase (SOD) and glutathione peroxidase (GPx) [50]. Animals lacking PGC1-α display reduced levels of SOD and GPx, whereas overexpressing PGC1-α in mouse leads to increased antioxidant defenses in muscles [50,51]. Therefore, reduced levels of PGC1-α in sarcopenia contribute to the disease pathogenesis also through an impairment of oxidative stress defenses.

Given the acquired jammed mitochondrial biogenesis observed in sarcopenia, mitochondrial quality control plays a pivotal role in mitochondrial homeostasis maintenance in this post-mitotic tissue. The mitochondrial quality control entails two key stress response pathways: the unfolded protein response of the mitochondria (UPR^mt^, coordinating recovery of challenged mitochondria) and the mitophagy (mediating removal and digestion of compromised mitochondria) [52]. Mitochondrial potential and transport, two altered features of sarcopenic muscles, are essential for mitochondrial quality control mechanisms; hence, UPR^mt^ and mitophagy are altered during pathological muscle aging [53,54] (Figure 2). Accordingly, numerous mitophagy and UPR^mt^ regulators decrease with age in mice and human sarcopenic muscles, including mitochondrial chaperons and components of the endo-phagosome assembly machinery [55,56,57]. This decline in mitophagy and UPR^mt^ efficiency during aging sarcopenia contributes to the progressive accumulation of dysfunctional organelles and further tissue damage. Importantly, muscle-specific inhibition of the mitophagy mediator, autophagy related 7 (Atg7), in mice leads to premature aging typified by oxidative stress, mitochondrial dysfunction, muscle loss, and weakness [57,58], underlining the importance of damaged mitochondrial removal for muscle function. Accordingly, mice lacking the mitochondrial protease, HtrA2/Omi, involved in the UPR^mt^ response, display a sarcopenia phenotype accompanied by an alteration of mitochondrial proteostasis and function [59]. Conversely, interventions aiming at improving UPR^mt^ and mitophagy mediate numerous anti-aging effects in different tissues from nematodes to mammals, including increased lifespan and fitness. Pharmacological-, genetic-, or exercise-induced mitochondrial quality control upregulation mediates a restoration of mitochondrial function and muscle homeostasis in aged flies, rodents, and patients [38,56,60,61,62,63]. The beneficial effects of mitochondrial quality control boosting in muscle tissue has encouraged the initiation of different clinical trials to assess the therapeutic potential of this strategy in muscle health and sarcopenic patients [64,65,66,67].

Mitochondrial dynamics are fundamental for correct mitochondrial quality control; hence, the proper balance between mitochondrial fusion and fission is critical for muscle homeostasis. Accordingly, mitochondrial dynamics and, consequently, mitochondrial morphology are profoundly altered in sarcopenia (Figure 2). Abnormalities in mitochondrial morphology observed in sarcopenia are a direct consequence of the dysregulation of key proteins involved in mitochondrial dynamics, including the fusion proteins, mitofusins 1 and 2 (MFN1 and MFN2) and optic atrophy protein 1 (OPA1), and the fission proteins, mitochondrial fission factor (MFF) and dynamin-related protein 1 (DRP1). In line, MFN2, OPA1, or DRP1 deficiency negatively affect muscle health and mitochondrial homeostasis, triggering the insurgence of sarcopenia features, such as muscle atrophy [68,69,70]. Importantly, sarcopenia patients also present altered expression of MFNs and DRP1, and consequent accumulation of dysfunctional and enlarged mitochondria [71]. Indeed, unbalanced fusion and fission leads in aged and sarcopenic muscle to the accumulation of swollen mitochondria that cannot be properly removed due to mitophagy flux impairment. This further exacerbates mitochondrial dysfunction by the inhibition of mitochondrial quality control and recycling, worsening tissue damage.

The concomitant stalling of mitochondrial biogenesis, reparation, and recycling ultimately provokes the accumulation of damaged and dysfunctional mitochondria in sarcopenic muscles. These defective organelles are a major source of cellular reactive oxygen species (ROS) (Figure 2). High metabolic tissues, such as skeletal muscles, process large quantities of oxygen; therefore, the loss of structural integrity of mitochondrial membranes and alteration of the respiratory machinery strongly promote ROS generation and oxidative damage. This accumulation of oxidative stress promotes consequent deterioration of muscle homeostasis [72]. Moreover, being the main source of ROS, mitochondria are also the primary targets of oxidative damage, which compromises mitochondrial quality control [73]. The equilibrium between ROS generation and scavenging is essential in maintaining cellular homeostasis. Hence, the increased level of oxidative stress observed in sarcopenia also reflects the inability of antioxidants to contain ROS overproduction. Indeed, despite an upregulation of some components of the cellular antioxidant defenses during sarcopenia due to compensatory mechanisms, their activity is not sufficient to prevent the accumulation of oxidative damage [74]. This is further exacerbated by reduced levels of PGC1-α, which modulates antioxidant enzymes synthesis, as previously mentioned. Interestingly, mice lacking Cu/Zn-superoxide dismutase (SOD1) display high levels of oxidative stress damage and early insurgence of sarcopenia [75,76]; similar results are observed when SOD1 ablation is restricted to muscle tissue [77]. Moreover, animals expressing muscle-specific mutated forms of SOD1 (SOD1^G93A^) showed neuromuscular junctions’ dismantlement, which is considered an early pathogenic signature of sarcopenia [78,79]. Conversely, strategies targeting oxidative stress have shown efficacy in preventing the development of the sarcopenic phenotype in aged rodents [80,81]; although, controversy still exist in the field due to different experimental designs [82,83].

## 3. Inflammageing

Aging is typically accompanied by a chronic state of low grade and sterile inflammation referred to as “inflammageing” (Figure 2). Inflammageing is associated with increased plasma levels of proinflammatory mediators, such as tumor necrosis factor α (TNFα), interleukin 6 (IL-6), and C-reactive protein (CRP). Increased circulating cytokines have been associated with the onset of different age-associated degenerative diseases, particularly muscle pathologies and concomitant reduced muscle mass and function. Indeed, elevated levels of proinflammatory cytokines are observed in diseases associated with muscle wasting, such as AIDS, chronic heart failure, chronic obstructive pulmonary disease, and cancer-related cachexia [84,85]. IL-6, TNFα, and CRP are also strongly upregulated in primary sarcopenia [86,87], suggesting a causal role of inflammageing in muscle loss.

A chronic state of inflammation in muscle tissue is often associated with altered cellular redox balance and oxidative stress. Increased ROS levels following mitochondrial dysfunction have been proposed as one of the main contributors in the development of sarcopenia-associated inflammageing, and of its impact in muscle function. In fact, nuclear factor kappa B (NF-κB), the main transcription factor involved in the induction of the inflammatory response in muscle tissue, can sense a cellular redox-state, allowing it, *de facto*, to be reactive towards ROS. Transgenic animals displaying constitutively activated NF-κB, indeed, are characterized by muscle wasting, whereas mice expressing a dysfunctional NF-κB are resistant to immobilization-induced muscle atrophy. Elevated oxidative stress influences NF-κB activity at different levels, by post-transcriptionally modulating upstream mediators of NF-κB response, or through promotion and stabilization of NF-κB binding to DNA. Stress-induced activation of NF-κB leads to upregulation of different cytokines, including IL-6 and TNF-α, which play a crucial role in the pathogenesis of sarcopenia. Increased levels of these cytokines further stimulate NF-κB activity and promote a chronic inflammatory state, triggering a pathological positive feedback loop, which drives muscle deterioration. In particular, TNF-α can mediate NF-κB upregulation, and mediate the loss of muscle proteins. Interestingly, this activation is redox sensitive, and the blockage of ROS production can prevent TNF-α mediated muscle wasting. Therefore, ROS play a crucial role in the development of inflammation in sarcopenia, and in mediating its detrimental effects by activating NF-κB either directly or indirectly.

Sarcopenia is characterized by a decrease in PGC-1α levels in muscle tissue, as mentioned above. This transcription factor plays a crucial role in inflammageing, and the inhibitory effects of PGC-1α on inflammatory response are the result of several cellular mechanisms. First, PGC-1α can directly modulate cytokines production by limiting NF-κB binding to DNA, hence reducing its detrimental effects on muscle tissue [88]. PGC-1α affects inflammation also by mediating the expression of different antioxidant enzymes [50]. Accordingly, PGC-1α overexpression in cultured muscle cells is sufficient to reduce the expression of inflammatory markers, including IL-6 and TNF-α [89]. Contrarily, muscle-specific ablation of PGC-1α leads to increased synthesis of these inflammatory cytokines in rodents and primary human muscle cells [89,90]. Interestingly, muscle-specific PGC-1α knockout mice also display reduced expression of antioxidant enzymes, suggesting also a role of PGC-1α in regulating inflammation through levels of ROS.

Inflammageing itself favours malnutrition and sarcopenia, as inflammatory cytokines activate protein catabolism and hormonal deregulation with increased cortisol production, which induces muscle waste and cachexia, and reduces anabolic hormones, such as growth hormones, sex hormones, and insulin growth factor (see [91] for a comprehensive review).

## 4. Malnutrition

The reduction of appetite and food intake is frequently associated with aging, and has been defined as “anorexia of aging” [92] (Figure 2). It is due to several conditions associated with aging itself, such as decreased salivation, difficulty swallowing, delayed emptying of the stomach and oesophagus, slower gastrointestinal movement [93], as well as a reduction of nutrient absorption capacity [94]. Other conditions associated with aging, such as drug use, loneliness, depression, lack of oral health, low quality of life, in addition to chronic non-communicable diseases, markedly increase the malnutrition risk [95]. Moreover, aging is associated with an imbalance in protein metabolism with increased catabolism, decreased anabolism, and reduced splanchnic extraction of amino acids [96,97]. Thus, both intrinsic and extrinsic factors contribute to malnutrition in old adults. Amongst extrinsic factors, hospitalization worsens malnutrition, affecting clinical outcomes, and resulting in increased morbidity and mortality [92,98]. Malnutrition is one of the risk factors predisposing to sarcopenia, and contributes to mitochondrial damage, an increase in inflammageing, and, consequently, to unhealthy and frail ageing [99,100]. Despite the importance of malnutrition, its reported incidence in older subjects varies, mainly because of the use of different definitions. It has been reported that malnutrition affected approximately 5 to 10% of community-dwelling older subjects; however, this percentage may rise up to 40% of hospitalized patients, and even up to 50% in patients in rehabilitation facilities [101].

A meta-analysis of whole protein supplements or high dietary protein intake trials have not been conclusive in terms of efficacy of this nutritional intervention [102,103]. Another meta-analysis shows that the efficacy of this nutritional strategy is only increased if combined with physical exercise [104]. The association between type and quality of diet beyond proteins, and the development of sarcopenia has been studied, and their meta-analysis showed that a diet rich in highly saturated fats, such as the Western diet, increases the risk of sarcopenia, whereas the Mediterranean and Nordic diets are associated with a lower risk [105].

Although protein-energy malnutrition is the most common form of malnutrition amongst older subjects, specific nutrient deficiencies, such as BCAAs, *n*-3 PUFAs, vitamin D, zinc, and selenium, play a major role in linking mitochondrial dysfunction, oxidative stress, inflammageing, and sarcopenia; hence, this review focuses specifically on these nutrients.

## 5. BCAAs

The aging process *per se* impairs protein metabolism, and favors muscle loss and the development of sarcopenia. BCAAs metabolism appears to be particularly impaired in older subjects, even though data on this topic are conflictual. Some studies suggest that there is a decreased blood availability of BCAAs in older subjects [106,107], whereas others do not [108,109].

Despite the controversy, several studies suggested that supplementation with a mixture of amino acids or essential amino acids might successfully counteract the development of sarcopenia by stimulating protein anabolism. A recent meta-analysis supports this hypothesis, showing that BCAAs-rich supplementation improves muscle mass and muscle strength in older subjects [36]. Similarly, previous meta-analyses suggest that essential amino acids are more effective in increasing muscle mass and function in old, malnourished patients, compared to non-essential amino acid or whole protein supplementations [110].

In aged animals, BCAAs-enriched supplements are effective in promoting mitochondrial formation and bioenergetics in skeletal muscles, with a consequent decrease in oxidative stress, and preservation of muscular function [111] (Figure 3). BCAAs target the mammalian site of the rapamycin (mTOR) signalling pathway, increase mitochondrial formation and nicotinamide adenine dinucleotide (NAD^+^) levels, and promote fatty acid oxidation, thus increasing energy production, and promoting muscle cellular homeostasis [111,112,113,114] (Figure 3). In addition, in vitro models confirm the ability of different BCAAs to boost mitochondrial activity [115,116,117,118]. We recently demonstrated that a BCAAs-enriched mixture is effective not only in increasing mitochondrial bioenergetics and mitochondrial function, and reducing oxidative stress in older malnourished patients, but also in rescuing the clinical sarcopenia phenotype [32] (Table 2). Different groups also showed the ability of BCAAs in decreasing inflammation in sarcopenic patients [119] (Table 2). Thus, the supplementation with BCAAs seems to be effective in clinical conditions characterized by increased protein catabolism, and, in particular, is able to influence mitochondrial function, and reduce oxidative stress in sarcopenia (Table 2).

A metabolite of leucine, beta-hydroxy-beta-methylbutyrate (HMB), has been shown to efficiently counteract protein catabolism and preserve muscle mass in old healthy adults submitted to bed rest [120]. Importantly, these effects on muscle structure are concomitant with a preservation of gene expression of mitochondrial genes. Moreover, HMB supplementation in combination with training rehabilitation (RT) leads to an increased in-muscle mitochondrial content, higher oxidative phosphorylation, and improved mitochondrial dynamics, effects not observed with RT alone [120]. HMB supplements are also capable of enhancing sarcolemma integrity, inhibiting protein degradation by stimulation of the ubiquitin pathway, increasing protein synthesis via the mTOR pathway, stimulating the growth hormone/insulin-like growth factor-1 (GH/IGF-1) axis, and enhancing muscle stem cell proliferation and differentiation [121].

## 6. Omega-3 PUFA

The link between muscle health and *n*-3 PUFA ingestion is well established [38]. In healthy middle-aged subjects, 8 weeks of 4 g supplementation increases the muscle protein fractional synthesis rate, the muscle protein concentration, and the protein/DNA ratio (i.e., muscle cell size) during insulin and amino acid infusion [122] (Table 2). At the molecular level, PUFA administration leads to increased synthesis of mitochondrial proteins in muscle tissue of older adults, with a concomitant reduction of oxidative stress and inflammation [123] (Table 2 and Figure 3). In line, a meta-analysis including 49 studies confirms that *n*-3 PUFA decreases the levels of IL-6 and CRP in middle-aged and older adults [124], thereby reducing the components of inflammageing, which contributes to sarcopenia. Further, *n*-3 PUFA has been shown to activate the mTOR and mitochondrial pathway in older adults, and to reduce insulin resistance (insulin being key to mTOR activations [125,126] (Figure 3)).

## 7. Vitamin D

Vitamin D deficiency or hypovitaminosis D is diagnosed by serum 25-hydroxyvitamin D (25(OH)D) levels of <50 nmol/L [127,128], and can be the consequence of different inappropriate lifestyles, such as malnutrition or reduced sunlight exposure [128,129,130]. Due to its diffusion, hypovitaminosis D has been defined as a pandemic. Prevalence increases with ageing due to different conditions, such as reduced cutaneous synthesis; lower sun daily exposure; or diseases, including chronic renal failure or gastrointestinal malabsorption [131,132]. Hypovitaminosis D has been associated with several chronic diseases, and is considered a causal factor for the development of sarcopenia and frailty [133,134]. In line, in animal models, diet-induced vitamin D deficiency leads to the manifestation of sarcopenia symptoms in skeletal muscles, including muscle weakness, decreased muscle force and physical performance, and reduced mitochondrial activity [129,135,136,137,138]. Moreover, reduced vitamin D levels have been associated with decreased muscular mass and strength in different human cohorts [139,140,141,142]. Despite these association studies and the biological plausibility of the link between vitamin D and muscle health, intervention studies are conflicting. In fact, some meta-analysis and systematic literature reviews have found only modest clinical effects of vitamin D supplements on muscle strength; these non-conclusive results are partially explained by the great inter-study heterogeneity regarding the population included, and the different molecules and doses used for the supplementation [143,144,145].

Several authors have studied the direct effect of vitamin D on mitochondrial function. Reduced vitamin D is associated with impaired mitochondrial function in skeletal muscle [146]. Moreover, vitamin D receptor (VDR) loss-of-function C2C12 myoblasts display severely compromised mitochondrial function [147], whereas muscle-specific VDR knock down in rodents leads to decreased expression of mitochondrial genes and sarcopenia [148,149]. On the other hand, 25(OH)D administration improves mitochondrial OXPHOS and dynamics in C2C12 myoblasts [146] (Figure 3). Accordingly, protective effects on mitochondrial functions of calcitriol, the active metabolite of vitamin D, and calcipotriol, an analogue of vitamin D, have been demonstrated in experimental models [150,151,152,153]. Despite the association and intervention studies in humans and experimental data suggesting a protective effect of vitamin D on mitochondrial function (Table 2), taken together, the available evidence is not sufficient to recommend treating patients affected by sarcopenia with vitamin D supplementation, even though they have proven hypovitaminosis D.

## 8. Selenium and Zinc

Selenium is an essential trace element that is cornerstone to the body’s antioxidant defence, being a structural component of the glutathione peroxidase (GPX) family of selenoenzymes. Deficiency is frequent in different parts of the world. Low plasma levels have been independently associated with poor muscle strength in community-dwelling older adults in Tuscany [154]. Moreover, selenium deficiency has been linked to numerous skeletal muscle disorders [155], including sarcopenia [156,157]. Selenium is found within muscles as selenocystein in selenoprotein N, which is involved in redox-modulated calcium homeostasis, and in protection against oxidative stress [158]. Accordingly, decreased levels of selenium are linked to impaired antioxidant defenses, and increased oxidative stress, with a consequent increase of muscle inflammation and mitochondrial abnormalities, including reduced copy number, increased size, altered cristae structure, and impaired respiration [159,160]. On the other hand, animal research has shown that selenium supplements improve muscle performance by modulating calcium metabolism and mitochondrial biogenesis [158,161] (Figure 3).

Zinc is a universal essential trace element, being the second most abundant element in the body after iron [162]. Its distribution in the body is not homogeneous, the majority (60%) being found in the muscle. In skeletal muscle, zinc has been shown to affect myogenesis and muscle regeneration due to its effects on muscle cell activation, proliferation, and differentiation [162]. Not surprisingly, zinc intake has been characterized as a predictor of reduced age-related skeletal muscle decline in older adults [163]. Among its multiple properties, zinc acts as an anti-inflammatory and anti-oxidant mediator, being a crucial component of the copper-zinc-dependent SOD (SOD1), involved in ROS scavenging and mitochondrial redox defenses [164], and by inhibiting NF-kB activation [165] (Figure 3). Moreover, *in vitro* zinc administration can induce mitophagy under oxidative stress condition in muscle cells, preventing ROS damage [166] (Figure 3). Zinc deficiency is widespread and particularly frequent in old adults [167]. Importantly, significantly decreased serum levels are observed in sarcopenic patients, underlining the role of this element in muscle homeostasis [168].

## 9. Conclusions

Sarcopenia is rapidly emerging as a global health concern due to its multifactorial nature, age-associated increase in prevalence, and unavailability of specific treatment. From a clinical point of view, sarcopenia is characterized by a progressive decrease in muscle mass, causing a deterioration in strength and physical performance, contributing to frailty and, ultimately, mortality. A major role in mediating these detrimental features of sarcopenia is played by mitochondrial dysfunction, and consequent oxidative stress and inflammageing. Malnutrition favors the development of sarcopenia, and has profound effects on mitochondrial function, cellular redox status, and inflammatory response. The role of different micro- and macro-nutrients in maintaining muscle health is well characterized; not surprisingly, they also play a role in preventing the onset of sarcopenia. Indeed, the administration of BCAAs, PUFA, vitamin D, zinc, and selenium is effective in ameliorating features of cellular senescence, namely mitochondrial homeostasis, oxidative stress, and inflammageing. Despite the beneficial effects observed on muscle-cell ageing suggesting a therapeutic role of these molecules in ameliorating sarcopenia, the evidence for promoting these nutrients for treating sarcopenia is still not sufficient. Carefully planned clinical studies of proper durations with correct doses, involving large numbers of patients, should be performed for these nutrients to translate from the bench to bedside.

## Figures and Tables

**Figure 1 nutrients-14-00483-f001:**
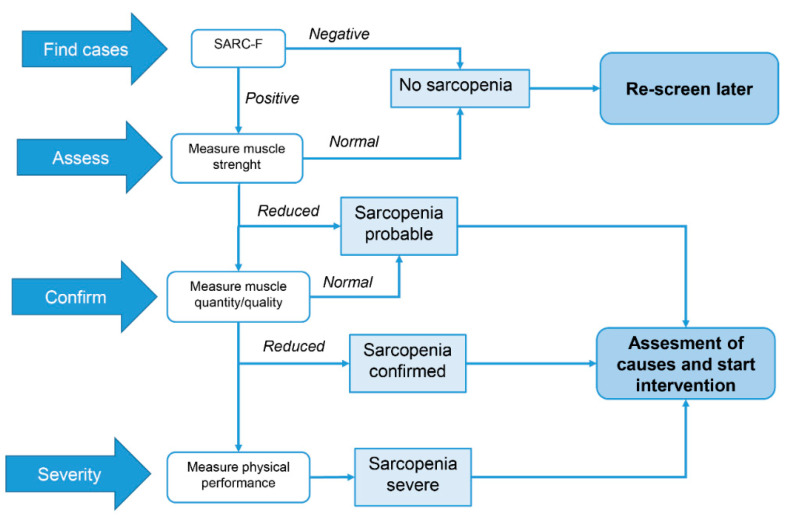
EWGSOP-2 algorithm for the diagnosis of sarcopenia. Find the case, make the diagnosis, and quantify the severity.

**Figure 2 nutrients-14-00483-f002:**
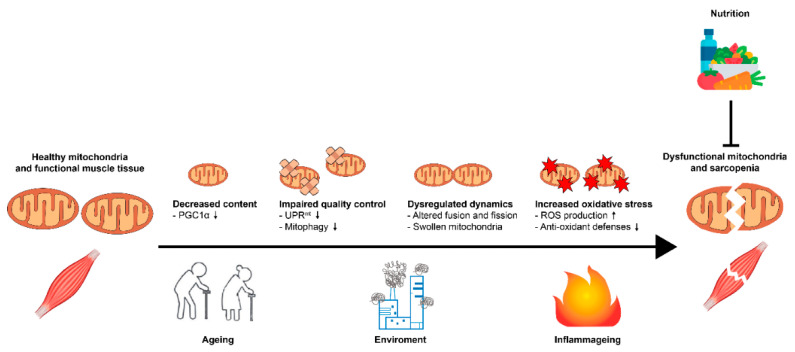
The relationship between mitochondrial dysfunction, oxidative stress, inflammageing, malnutrition, and sarcopenia. ↓: decreased; ↑: increased.

**Figure 3 nutrients-14-00483-f003:**
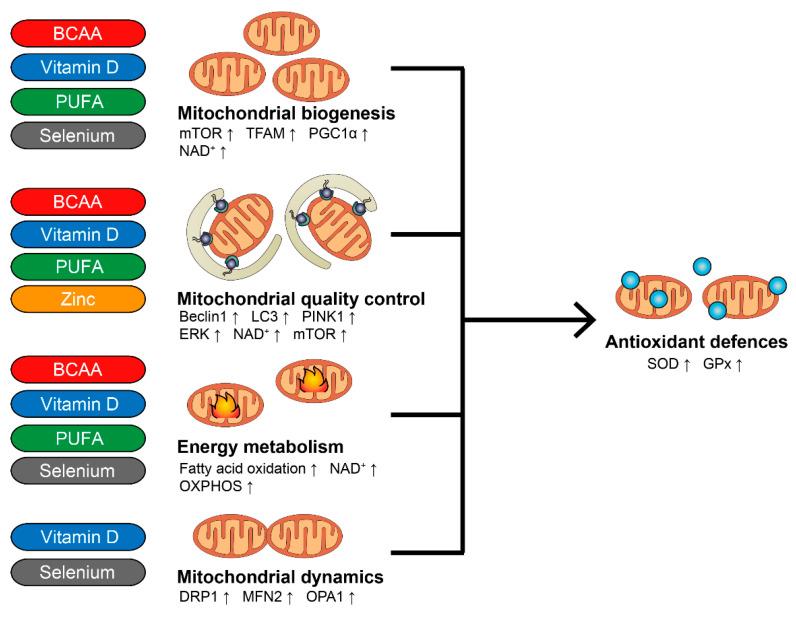
Mitochondrial pathways and molecular mediators affected by BCAA (in red), vitamin D (in blue), PUFA (in green), zinc (in orange), and selenium (in grey). ↑: increased.

**Table 1 nutrients-14-00483-t001:** Different criteria for the diagnosis of sarcopenia.

Criteria	Muscle Performance	Muscle Strenght	Lean Body Mass	Summary Definition
International Working Group [7]	Gait speed < 1.0 m/s	Not included	ALM/ht^2^ ≤ 7.23 kg/m^2^	Sarcopenia: slowness and low lean mass
EWGSOP-1 [8]	Gait speed ≤ 0.8 m/s	Grip strength < 30 kg	2 SD < mean reference value	Sarcopenia: low lean mass and slowness or weakness Severe sarcopenia: all three criteria
EWSGOP-2 [10]	Gait speed ≤ 0.8 m/s SPPB ≤8 point score TUG ≥ 20 s 400 m walk test ≥6 min	Grip strength <27 kg for men and <16 kg for women Chair stand for five rises > 15 s	ALM/ht^2^ < 7.0 kg/m^2^ for men ALM/ht^2^ < 5.5 kg/m^2^ for women	Sarcopenia: weakness and low lean mass Severe sarcopenia: all three criteria
FNIH [9]	Gait speed ≤ 0.8 m/s	Grip strength < 26 kg	ALM/BMI < 0.789	Sarcopenia: weakness and low lean mass Severe sarcopenia: all three criteria
Baumgartner [6]	Not included	Not included	ALM/ht^2^ ≤ 7.23 kg/m^2^	Low lean mass
Newman [5]	Not included	Not included	Residual of actual ALM-predicted ALM from equation	Low lean mass

ALM: appendicular lean mass; SD: standard deviation; SPPB: short performance physical battery; TUG: timed up and go test.

**Table 2 nutrients-14-00483-t002:** Effects of dietary intervention on mitochondrial function, oxidative stress level, and clinical features of sarcopenia.

Dose	Subjects	Mitochondria	Muscle	Study Design and References
BCAAs				
Leucine (1250 mg), Lysine (650 mg), Isoleucine (625 mg), Valine (625 mg), Threonine (350 mg), Cystine (150 mg), Histidine (150 mg), Phenylalanine (10 mg), Methionine (50 mg), Tyrosine (30 mg), Tryptophan (20 mg), Vitamin B 6 (0.1 mg), Vitamin B1 (0.15 mg). Twice a day for 2 months.	116 men and women aged 80 years or older.	ATP ↑ Electron flux ↑ Fusion ↑ Oxidative stress ↓ *	Strength ↑ Walking distance ↑ Balance ↑ Risk of falls ↓ Protein synthesis ↑ Lean mass ↑ Insulin sensitivity ↑	Randomized Controlled Trial [32]
l-leucine (2.5 g), l-lysine (1.3 g), l-isoleucine (1.25 g), l-valine (1.25 g), l-threonine (0.7 g), l-cysteine (0.3 g), l-histidine (0.3 g), l-phenylalanine (0.2 g), l-methionine (0.1 g), l-thyrosine, (0.06 g), l-tryptophan (0.04 g). Twice a day for 8 months	41 men and women aged 66–84 years with diagnosed sarcopenia.		TNFα ↓ Lean mass ↑ Insulin sensitivity ↑	Randomized Controlled Trial [118]
Histidine (0.82 g), Isoleucine (0.78 g), Leucine (1.39 g), Lysine (1.17 g), Methionine (0.23 g), Phenylalanine (1.17 g), Threonine (1.10 g), Valine (0.86 g). Twice a day for 3 months.	14 women aged 68 +/− 2 years.		Fractional synthesis rate ↑ Lean mass ↑	Randomized, Controlled Trial [169]
(mg · mL^−1^ and (mmol · l^−1^), respectively): Alanine 20.7 (232.3), arginine 11.5 (66.0), glycine 10.3 (137.2), histidine 4.8 (30.9), isoleucine 6.0 (45.7), leucine 7.3 (55.6), lysine 5.8 (39.7), methionine 4 (26.8), phenylalanine 5.6 (33.9), proline 6.8 (59.1), serine 5.0 (47.6), threonine 4.2 (35.3), tryptophan 1.8 (8.8), tyrosine 0.4 (2.2), and valine 5.8 (49.5). The total amino acid infusion was 148.5 mg × kg^−1^ × h^−1^ for 480 min.	5 subjects aged 71+/− 2 years.		Fractional synthesis rate ↑	Longitudinal Clinical Trial [170]
L-leucine (1.3 g), L-lysine (0.66 g), L-isoleucine (0.6 g), L-valine (0.63 g), L-threonine (0.36 g), L-cystine (0.13 g), L-histidine (0.13 g), L-phenylalanine (0.1 g), L-methionine (0.06 g), L-tyrosine (0.03 g), L-triptophane (0.03 g). 3 times a day for 3 months.	One hundred men and women aged >65 years.		Strength ↑ Walking distance ↑ Myocardial performance ↑	Randomized Controlled Trial [171]
Omega-3 PUFA				
Ethylesters of eicosapentaenoic acid (1.86 g), docosahexaenoic acid (1.50 g). Once a day for 8 weeks.	5 men and 4 women aged 25–45 years		Protein concentration ↑ Cell size ↑	Longitudinal Clinical Trial [122,172]
EPA (1.35 g), DHA (0.6 g). Twice a day for 4 months.	12 young (18–35 years) and 12 older (65–85 years) men and women.	Biogenesis ↑ Oxidative stress ↓ **	Fractional synthesis rate↑	Longitudinal Clinical Trial [123]
EPA (0.72 g), DHA (0.24 g). Twice a day for 6 months.	126 women aged between 64–95 years.		Walking speed ↑	Randomized Controlled Trial [173]
EPA (0.93 g), DHA (0.75 g). Twice a day for 6 months.	60 men and women aged 60–85 years.		Thigh muscle volume ↑ Handgrip strength ↑ Upper & lower-body muscle strength ↑	Randomized Controlled Trial [174]
Vitamin D				
Vitamin D3 (0.5 mg) on alternate days for 3 months.	12 individuals with severe vitamin D deficiency aged 18.1–50.4 years and 15 age-matched controls.	OXPHOS ↑ **	Fatigue ↓ Phosphocreatine recovery half-time ↓	Longitudinal Clinical trial [175]
Vitamin D3 (60,000 IU/week) for 3 months.	16 females and 3 males, mean age 17–24 years.	ATP ↑ **	Pi:PCr ↑	Longitudinal Clinical Trial [176]

OXPHOS: Oxidative phosphorylation; ATP: adenosine triphosphate; Pi:PCr: inorganic phosphate to phosphocreatine ratio. ↓: decreased; ↑: increased. * mitochondria analyzed in PBMC; ** mitochondria analyzed in muscle tissue.

## Data Availability

Not applicable.

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
