# Peer review of "From the Bench to the Bedside: Branched Amino Acid and Micronutrient Strategies to Improve Mitochondrial Dysfunction Leading to Sarcopenia"

_nutrients, 2022, doi:10.3390/nu14030483_

Round 1
Reviewer 1 Report
Tittle:
From the bench to the bedside: enhanced dietary strategies to delay cellular senescence leading to sarcopenia.
Version: December 31th, 2021.
Nutrients. MDPI
Reviewer's report:
This is a review article with aiming to discuss molecular features of cellular senescence leading to sarcopenia, namely impaired mitochondrial homeostasis, increased oxidative stress and chronic inflammaging. Furthermore authors describe the effects of nutrients, such as BCAAs, n-3 PUFAs, vitamin D, zinc and selenium in mitochondrial dysfunction, oxidative stress, inflammaging and sarcopenia. It could be of interest to investigators and clinicians. The topic of the manuscript is appropriate for the Journal. However, minor essential revisions are necessary.
Minor essential revisions
One doubt about this manuscript is why authors included “we discuss molecular features of cellular senescence leading to sarcopenia, namely impaired mitochondrial homeostasis, increased oxidative stress and chronic inflammaging”. Cellular senescence is the consequence of wide range of stressors contribute to sarcopenia. Authors only mentioned three molecular pathways. Could clarify this. Please check:
• Mankhong S, Kim S, Moon S, Kwak HB, Park DH, Kang JH. Experimental Models of Sarcopenia: Bridging Molecular Mechanism and Therapeutic Strategy. Cells. 2020;9(6):1385. Published 2020 Jun 2. doi:10.3390/cells9061385
For example, check:
Van Deursen, J.M. The role of senescent cells in ageing. Nature 2014, 509, 439–446.
A wide range of stressors, including DNA damage, shortening of telomere length, oxidative stress, mitochondrial dysfunction, oncogenic activation, and chemotherapeutic agents drives cellular senescence. Why is the reason for including only three? Other mechanisms are important too for developing sarcopenia. What is exactly the focus of this review. Please clarify.
For example, in reference 29 … Other senescent cell markers include activated and persistent DNA-damage response (see “DNA repair”), telomere shortening and dysfunction (see “Telomere”), and senescence-associated secretory phenotype (SASP) (see “Inflammation and intercellular communication”). Tittle insist in “delay cellular senescence”. So, please clarify these no inclusion of this key information about cellular senescence in the main text.
Why authors non mentioned other several fundamental mechanisms of aging that contribute to sarcopenia as satellite cell dysfunction, protein degradation or protein synthesis? Please clarify this situation. Please check figure 1 of Mankhong et al. paper referred above.
In the same way, another question, authors mentioned in their tittle “dietary strategies” … why any mention of another metabolism pathways as nutrient sensing, protein or lipids metabolism? Please any mention is strong recommended about possible biomarkers of sarcopenia related with metabolism other than oxidative stress, mitochondrial homeostasis and inflammaging.
Why abbreviations are no clarified? For example. Pag 5 Line 108 PCG1 alfa
The abstract is concise and specific. However should highlight only the mechanisms included in discussion.
Furthermore, pl check for the tittle the comment above about cellular senescence or clarify this in the main text.
Thanks for letting me review this manuscript.
This could be a nice paper.
Level of interest: An article whose findings are important to those with closely related research interests.
Quality of written English: Well.
Statistical review: No.
Declaration of competing interests:
I declare that I have no competing interest.
Author Response
The authors gratefully thank the reviewer for his/her useful comments.
Reviewer: One doubt about this manuscript is why authors included “we discuss molecular features of cellular senescence leading to sarcopenia, namely impaired mitochondrial homeostasis, increased oxidative stress and chronic inflammaging”. Cellular senescence is the consequence of wide range of stressors contribute to sarcopenia. Authors only mentioned three molecular pathways. Could clarify this.
Authors: Thank you for your comments, the reason why we chose only three molecular pathways leading to sarcopenia is now discussed in the text (lines 78-90) and suggested articles have been added in the references.
Reviewer: In the same way, another question, authors mentioned in their tittle “dietary strategies” … why any mention of another metabolism pathways as nutrient sensing, protein or lipids metabolism? Please any mention is strong recommended about possible biomarkers of sarcopenia related with metabolism other than oxidative stress, mitochondrial homeostasis and inflammaging.
Authors: literature of whole proteins and dietary patterns have been added in the “malnutrition section”.
Reviewer: Why abbreviations are no clarified? For example. Pag 5 Line 108 PCG1 alfa
Authors: Thank you for your comment, abbreviations are now clarified and written out.
Reviewer: The abstract is concise and specific. However should highlight only the mechanisms included in discussion.
Authors: The abstract has been reworded according to the reviewer’s suggestion.
Reviewer: Furthermore, pl check for the tittle the comment above about cellular senescence or clarify this in the main text.
Authors: The review title has been changed into: “From the bench to the bedside: enhanced dietary strategies to improve mitochondrial dysfunction leading to sarcopenia”.
Reviewer 2 Report
Comments to the Author
This is a practically important review article indicates that From the bench to the bedside: enhanced dietary strategies to delay cellular senescence leading to sarcopenia.
However, it is necessary to reexamine the research method etc. in several respects.
1. Table 1. Different criteria for the diagnosis of sarcopenia.
EWGSOP-2 cut-off points
ALM/ht2<7.0kg/m2 for men
ALM/ht2<5.5kg/m2 for women
Age and Ageing 2019; 48: 16–31. doi: 10.1093/ageing/afy169
EWGSOP-1 cut-off points
There is no mention of an ALM/ht2 and grip strength cut-off points for men and women in Reference 8.
2. Please show graphically the pathways of vitamin D, PUMA, BCAA, Se, Zn intake and muscle protein metabolism in sarcopenia. In particular, previous reviews have been unclear about the pathways of PUMA and Se, Zn intake and muscle protein metabolism in sarcopenia.
3. Table 2. Effects of dietary intervention on muscle mitochondrial function, oxidative stress level and clinical features of sarcopenia.
Please provide a detailed summary of the clinical features of muscle mitochondrial function, oxidative stress levels and sarcopenia with dietary intervention.
For example, summarize dose, frequency of intake, duration of intake, subjects, age, gender, exercise intervention, study design (cross-sectional studies, cohort studies, randomized controlled trials, etc.) and outcomes (mitochondria and muscle tissue) in each reference.
“Mass↑” was lean mass?, body mass?
4. As a review article, I think the novelty is unclear. Please emphasize the novelty. Several reviews have been reported, especially on BCAA. 
Author Response
The authors gratefully thanked the reviewer for his/her useful comments.
Reviewer: Table 1. Different criteria for the diagnosis of sarcopenia.
Authors: The table has been corrected, thank you.
Reviewer: Please show graphically the pathways of vitamin D, PUMA, BCAA, Se, Zn intake and muscle protein metabolism in sarcopenia. In particular, previous reviews have been unclear about the pathways of PUMA and Se, Zn intake and muscle protein metabolism in sarcopenia.
Authors: A new Figure 3 has been designed according to reviewer’s suggestion and is now added in the manuscript.
Reviewer: Table 2. Effects of dietary intervention on muscle mitochondrial function, oxidative stress level and clinical features of sarcopenia. Please provide a detailed summary of the clinical features of muscle mitochondrial function, oxidative stress levels and sarcopenia with dietary intervention.
Authors: The table has been amended according to the referee’s suggestion.
Reviewer:4. As a review article, I think the novelty is unclear. Please emphasize the novelty. Several reviews have been reported, especially on BCAA.
Author: A paragraph highlighting the novelty has been added in the introduction section (lines 97-101).
Round 2
Reviewer 2 Report
Several issues of the manuscript have been appropriately corrected by revision, so I think that it is appropriate for publication.